# Assessing the healthiness of UK food companies' product portfolios using food sales and nutrient composition data

**Lauren Kate Bandy**[1]*, **Sven Hollowell**[1], **Richard Harrington**[2], **Peter Scarborough**[1], **Susan Jebb**[3], **Mike Rayner**[1]

**1** Nuffield Department of Population Health, University of Oxford, Oxford, England, **2** Nuffield Department of Population Health, Centre on Population Approaches for NCD Prevention, University of Oxford, Oxford, England, **3** Nuffield Department of Primary Care Health Sciences, Radcliffe Observatory Quarter, University of Oxford, Oxford, England

* lauren.bandy@ndph.ox.ac.uk

**Data Availability Statement:** This study used data from two commercial sources. The sales data was accessed under licence from Euromonitor International (https://www.euromonitor.com/

## Abstract

### Background

The provision and over-consumption of foods high in energy, saturated fat, free sugars or salt are important risk factors for poor diet and ill-health. In the UK, policies seek to drive improvement through voluntary reformulation of single nutrients in key food groups. There has been little consideration of the overall progress by individual companies. This study assesses recent changes in the nutrient profile of brands and products sold by the top 10 food and beverage companies in the UK.

### Methods

The FSA/Ofcom nutrient profile model was applied to the nutrient composition data for all products manufactured by the top 10 food and beverage companies and weighted by volume sales. The mean nutrient profiling score, on a scale of 1–100 with thresholds for healthy products being 62 for foods and 68 for drinks, was used to rank companies and food categories between 2015 and 2018, and to calculate the proportion of individual products and sales that are considered by the UK Government to be healthy.

### Results

Between 2015 and 2018 there was little change in the sales-weighted nutrient profiling score of the top 10 companies (49 to 51; p = 0.28) or the proportion of products classified as healthy (46% to 48%; p = 0.23). Of the top five brands sold by each of the ten companies, only six brands among ten companies improved their nutrient profiling score by 20% or more. The proportion of total volume sales classified as healthy increased from 44% to 51% (p = 0.07) driven by an increase in the volume sales of bottled water, low/no calorie carbonates and juices, but after removing soft drinks, the proportion of foods classified as healthy decreased from 7% to 6% (p = 33).

packaged-food) via the Bodleian Library, University of Oxford, using Euromonitor's database portal Passport GMID. The product information dataset, including nutrition composition data, was purchased for the purpose of the lead author's DPhil research project from Edge by Ascential (https://www.ascentialedge.com/our-solutions). Due to licencing restrictions, the Euromonitor and Edge by Ascential datasets can only be requested under licence for the purpose of verification and replication of study's findings via the research group's Data Access Committee (contact: Trisha Gordon foodDBaccess@ndph.ox.ac.uk). Further use of these datasets must be negotiated with the data owners (Euromonitor contact: Ashton Moses - passport.support@euromonitor.com, Edge by Ascential contact: David Beech - info@ascentialedge.com). The authors received no special privileges in accessing the data.

**Funding:** LB, SH and MR are funded by the Nuffield Department of Population Health, University of Oxford. PS is funded by a British Heart Foundation Intermediate Basic Science Research Fellowship (FS/15/34/31656). All authors are part of the National Institute for Health Research (NIHR) Oxford Biomedical Research Centre (BRC). SJ is also funded by the NIHR Collaboration for Leadership in Applied Health Research and Care Oxford at Oxford Health NHS Foundation Trust and is an NIHR senior investigator. The funders had no role in study design, data collection and analysis, decision to publish, or preparation of the manuscript.

**Competing interests:** The authors have declared that no competing interests exist.

## Conclusions

The UK voluntary reformulation policies, setting targets for reductions in calories, sugar and salt, do not appear to have led to significant changes in the nutritional quality of foods, though there has been progress in soft drinks where the soft drink industry levy also applies. Further policy action is needed to incentivise companies to make more substantive changes in product composition to support consumers to achieve a healthier diet.

## Introduction

The provision and consumption of foods high in energy, saturated fat, free sugars or salt is an important marker of poor diet and associated with substantial morbidity [1]. To support improvements in public health nutrition, Public Health England (PHE) published a series of voluntary, category-specific reformulation targets for calories, sugar and salt [2–4] to encourage manufacturers to improve the nutritional quality of everyday products. Progress has been monitored by measuring change in the levels of individual nutrients and does not include a more holistic view of how the nutritional quality of products has changed overall.

The food industry in the UK is powerful and consolidated; in 2018, the retail value sales of packaged food and soft drinks products was £71.3 billion, with the 10 largest companies accounting for nearly a quarter (24%) of the total [5]. In order for PHE's voluntary reformulation targets to be successful in improving quality of the UK population's diet, food manufacturers–especially the largest companies whose products dominate the market—must make changes across a range of products. So far, PHE has focused on changes in specific food groups and has published only limited company-level analysis, but progress by company is vital to understanding the industry response to the targets.

Nutrient profiling is "the science of classifying and ranking foods according to their nutritional composition for reasons related to preventing disease and promoting health" [6]. Nutrient profiling generally involves the application of a model that classifies or ranks foods based on their overall nutrition composition, rather than looking at individual nutrients in isolation. It has multiple purposes, including supporting health-related labelling schemes and restricting the marketing of foods to children [7]. The UK Government's current nutrient profile model was developed by the Food Standards Agency (FSA) to provide the Office for Communications (Ofcom) with a tool to differentiate between foods that can and cannot be advertised to children, based on their nutrition composition [8].

The aim of this study was to assess how the nutritional quality of products offered by the top 10 global food and drink companies has changed over time by applying the FSA/Ofcom nutrient profiling model to a composition database, and weighting it using product sales data.

## Methods

### Data types and sources

Volume sales data was sourced from Euromonitor and accessed through the Oxford University Library. The top 10 UK food and soft drink manufacturers and their brands were identified based on global company names using 2018 sales data from Euromonitor [5]. A company is defined by Euromonitor as: "the legal entity that produces or distributes an individual or group of brands in the UK". All of the brands manufactured by these companies between 2015 and 2018 were identified, including those that dropped in or out of the market. Brands were

defined as a set of products that have the same generic name and are manufactured by one company.

The composition data were provided by Edge by Ascential (previously Brand View), a private analytics company that collects product information, including nutrient composition data, by scraping the websites of the UK's three leading retailers: Asda, Sainsbury's and Tesco. These data were scraped from these three websites on the same date (13th December) for four consecutive years (2015, 2016, 2017 and 2018). The sales data and nutrition composition data were automatically matched in Python based on three identifier variables that were present in both databases: brand name, category and year. A 10% random sample of brands was checked manually for any errors. Of the 20 brands checked, 4 brands were identified as pairing with the correct brand name but incorrect category. All 4 of these errors were brands that appeared in more than one category (e.g. Cadbury is present in five categories, including baked goods and confectionery). The matching code was adjusted so that it first paired based on matching categories, and then brand names, and no errors were identified after further checks.

## Applying the FSA/Ofcom nutrient profile model

The FSA/Ofcom nutrient profile model was applied to the individual product composition data. The appropriate points were awarded based on each product's energy, saturated fat, total sugar and sodium content ("A-points") and fibre, protein and fruit, nut and vegetable (FNV) content ("C-points") per 100g, as set out by FSA/Ofcom's technical guidance [9]. This system was developed for the purposes of restricting advertising of food to children, but here we have used it to classify products as healthy and unhealthy. A food is classified as 'less healthy' if it scores four points or more. A drink is classified as 'less healthy' if it scores 1 point or more. For the purpose of comparing companies' entire product portfolios, we converted the nutrient profile score to a 1–100 scale (-2(original score) +70), so that a higher score indicates healthier products. In order to directly compare drink scores with food scores, we also applied a linear adjustment to the distribution of the soft drinks scores (11x – 704, where x is the score for drinks on the 1–100 scale). The linear adjustment was selected so that the 33rd percentile and 66th percentile of both foods and drinks received the same score (44 and 66, respectively). After the scale conversion and linear adjustment, the thresholds for products to be considered healthy according to the FSA/Ofcom nutrient profile model were 62 or more for foods and 66 or more for drinks.

If the nutrient content for a product was missing, then data was imputed by calculating a brand average for foods in the same category, and if this was not possible, an overall category average. FNV content was estimated based on the ingredients list to categorise ingredients into 'fruit', 'nut', 'vegetable' and 'other'. The percentage composition of ingredients was identified if this information was provided in the ingredients list. For the products where percentage of ingredients were not given, values were imputed based on a brand and category average, or if this was not possible, a category average.

## Variables calculated

The total value (£ millions) and volume of food and soft drinks (tonnes) and the sales weighted mean nutrient profiling score (referred to in figure labels as sales-weighted score) were calculated in R for each company and brand, both overall and by category. When one brand had multiple product variants, a simple mean was used. While all brands were included in the analysis, only the top five for each company (n = 50) were presented for the brand-level analyses (Fig 3) for clarity. Bubble and chewing gum and milk formulas for infants, toddlers and children were excluded.

### Statistical analysis

Chi-squared tests were performed in R to test if there were any significant changes in the number of brands and products each company manufactured over time (2015–2018). ANOVA tests were used to test for differences over time in the nutrient profiling scores overall and for each company, category and brand.

## Results

In 2018, the top 10 food and soft drink companies had total value sales of £17.1 billion (**Table 1**). The top 10 companies by value were also the largest 10 in terms of volume sales, although there is variation in the ranking between these two measures. Food company Mondelez is the largest in value terms, while Coca Cola is the largest company in volume terms.

In 2018, there were 3273 individual products produced by these companies and included in the dataset under 222 different brands. Premier Foods had the largest product portfolio in 2018, with 613 individual products. Kellogg had the smallest, with 91 individual products. There was a decline in the total number of products that were manufactured by the top 10 companies over the period of analysis, from 3471 in 2015 to 3273 in 2018, a reduction of 6% (p <0.05). Seven out of ten of the companies reduced the number of products they manufacture.

Between 2015 and 2018 there was little change in the sales-weighted mean nutrient profiling score of all the products manufactured by included companies, moving from 49 to 51 (p = 0.28). The number of individual products that could be classified as healthy also remained relatively unchanged, at 46% in 2015 and 48% in 2018 (p = 0.23) There was an increase from 44% to 51% in the total volume sales classified as healthy (p = 0.07). Once soft drinks were removed, the proportion of volume sales that were classified as healthy decreased from 7% in 2015 to 6% in 2018 (p = 0.33).

The company that saw the largest increase in sales-weighted nutrient profiling score was Coca-Cola (48 to 51), although its score still remained below the FSA/Ofcom threshold (**Fig 1**). The company with the highest sales-weighted nutrient profiling score was Danone, with a large proportion of sales from dairy and bottled water, followed by Kraft Heinz, which has high volume sales of high-scoring pre-prepared baby foods. Coca-Cola, Mars, Unilever, Nestlé and Mondelez scored poorly, with portfolios dominated by confectionery and snacks.

Baby food had the healthiest nutrient profiling score in 2018, at 72 (**Fig 2**) but little change over time. Spreads, confectionery and ice cream and desserts were the categories with the

**Table 1. Number of products, brands and total volume sales by company, 2018.**

| Company Name | Value sales (£mn) | Equivalent value sales per person per day (£) | Total volume sales ('000 tonnes) |
|---|---|---|---|
| **Mondelez** | 2903 | 0.12 | 286 |
| **PepsiCo** | 2541 | 0.10 | 1073 |
| **Mars** | 2228 | 0.09 | 155 |
| **Coca-Cola** | 2167 | 0.09 | 1948 |
| **Nestlé** | 1531 | 0.06 | 92 |
| **Danone** | 1418 | 0.06 | 108 |
| **Premier Foods** | 1346 | 0.06 | 74 |
| **Unilever** | 1160 | 0.05 | 185 |
| **Kraft Heinz** | 930 | 0.04 | 61 |
| **Kellogg** | 858 | 0.04 | 144 |
| **Total** | **17,081** | **0.70** | **4,126** |

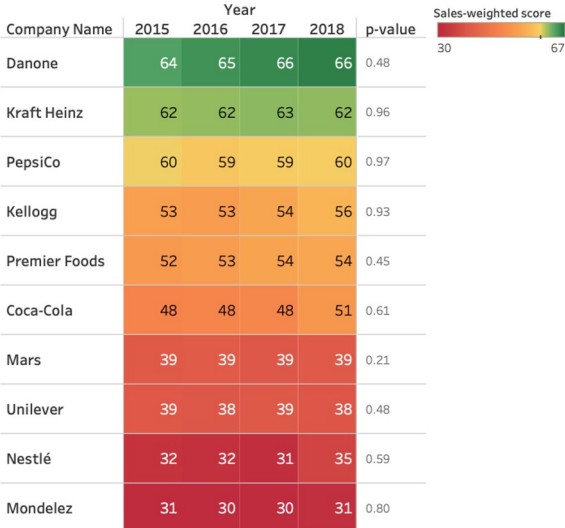

**Fig 1. Total sales-weighted nutrient profiling score by company and year.**

lowest nutrient profiling score. There was weak evidence of increases in score over time of staples, dairy, soft drinks and baked goods.

There was great heterogeneity between companies within some categories (**Fig 3**). For example, the company scores within the baked goods category ranged from 22 (Nestlé) to 69 (Premier Foods). In contrast, there was less variation within savoury snacks (39–52) and confectionery (26–42). Coca-Cola was the least diverse company producing only soft drinks, while Mondelez and Nestlé were the most diverse, with their portfolios containing products from six categories.

Of the five top-selling brands of each company, there were increases in the sales-weighted nutrient profiling score over time for Fanta (Coca-Cola), Volvic (Danone), San Pellegrino (Nestlé), Coco-Pops (Kellogg), Maltesers (Mars) and Angel Delight (Premier Foods) (**Fig 4**).

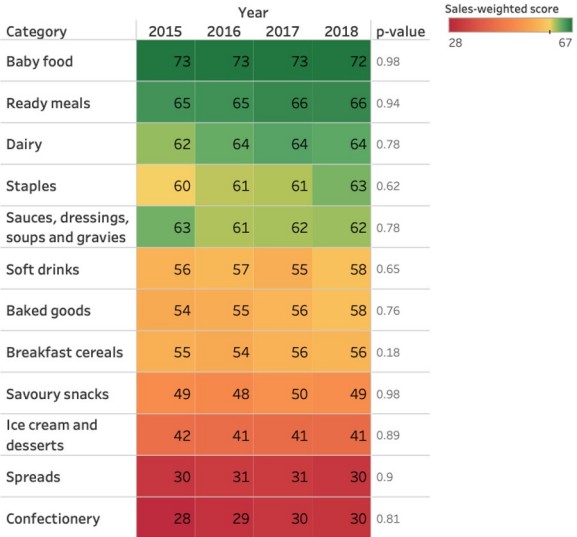

**Fig 2. Total sales-weighted nutrient profiling score by category and year.**

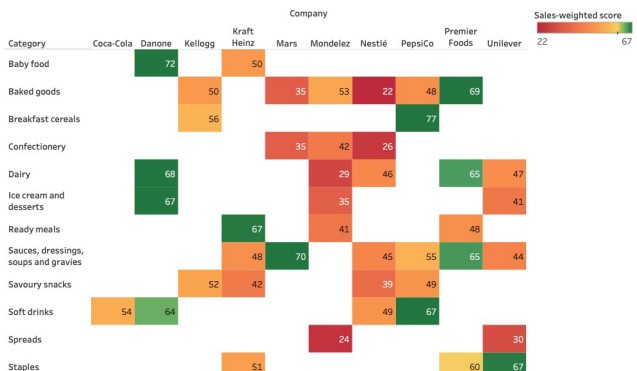

**Fig 3. Sales-weighted nutrient profiling score by company and category, 2018.**

Only Special K (Kellogg) saw its score cross the Ofcom threshold, up from 58 in 2015 to 62 in 2018 (+7%, p = 0.10). The largest increases were seen in soft drink brands San Pellegrino (+88%, p<0.01), Fanta (+28%, p<0.01) and Volvic (+26%, p<0.01) due to reductions in sugar and energy content. Tropicana (PepsiCo) saw a significant decrease in its score (-14%, p<0.01) due to a reduction in the proportion of sales of reduced sugar products, where the number of different products decreased over time. Coco-Pops (Kellogg) improved its score with an increase of 27% (p<0.01) due to a reduction in sugar, energy and salt. There was no strong evidence for changes in the scores of the top 5 brands for Kraft Heinz, Mondelez, PepsiCo and Unilever.

## Discussion

Between 2015 and 2018, there was no evidence of change in the overall mean sales weighted nutrient profiling score of products sold by the top 10 food and drink companies in the UK. This mean score remained well below the Ofcom threshold for broadcast advertising. There was only one company (Kellogg's) where there was weak evidence for improvement in its overall company score due to reductions in sugar and salt in two of its leading brands (Coco-Pops and Special K). There was a very small increase in the number of products classified as healthy (46% in 2015 to 47% in 2018) but a greater increase in the proportion of sales that were classified as healthy (44% in 2015 to 51% in 2018). This was largely attributable to a reduction in the sugar content of some soft drink products and an increase in the volume sales of healthy beverages (bottled water, low/no calorie drinks and fruit juices), changes likely driven by the introduction of the Soft Drink Industry Levy in 2018 [10,11]. Once soft drinks were removed, the proportion of healthy sales fell to 6% in 2018, down from 7% in 2015. This suggests that despite PHE's reformulation targets for calories, sugar and salt, there has been no improvement in the nutritional quality of foods that people are buying.

### Strengths and limitations

By pairing composition data with sales data and applying a nutrient profile model, both the relative healthiness of individual foods and drinks available, and the relative healthiness of what is sold have been assessed, and how this has changed over time. This gives an idea of how companies are responding to voluntary reformulation targets to improve the nutritional quality of their products overall, rather than in relation to a single nutrient.

Only 10 companies, based on global company name, were included in the analysis, which represented 24% of total value sales in the UK in 2018 [5]. These companies were selected

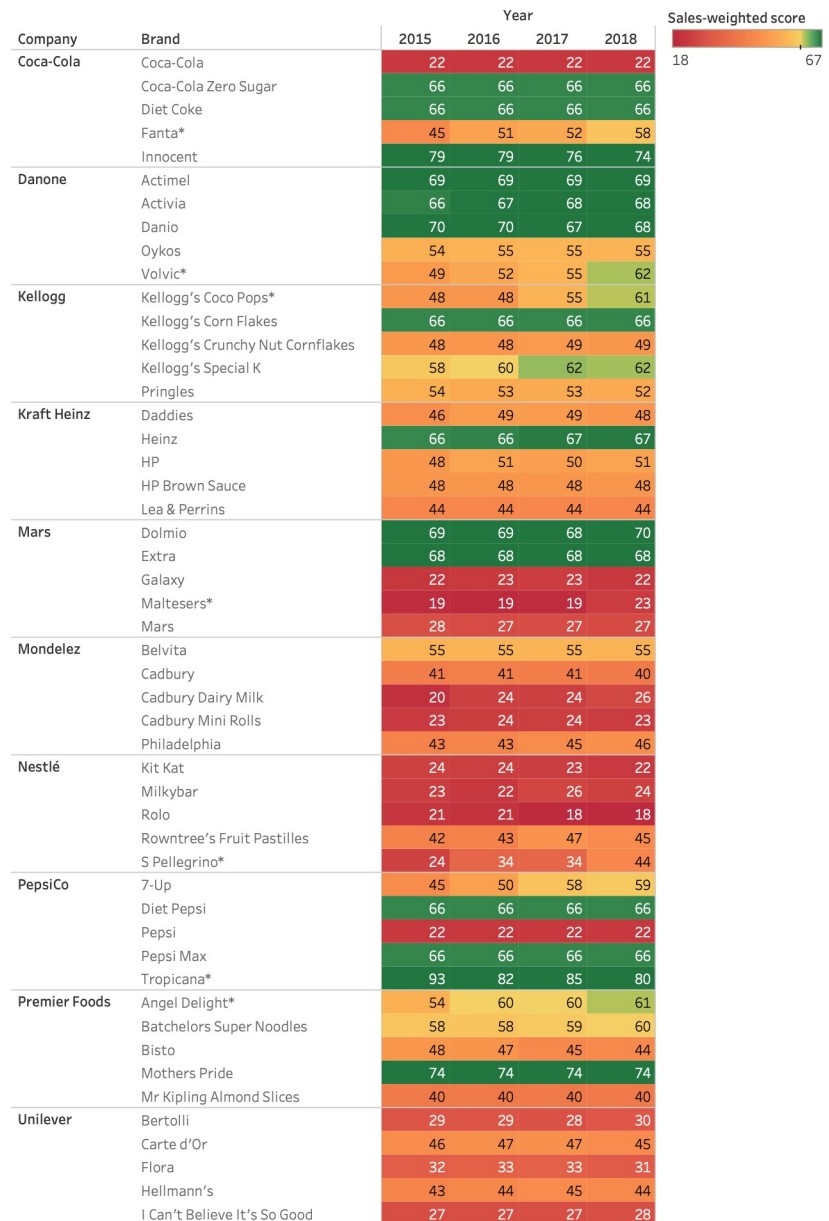

**Fig 4. Sales-weighted nutrient profiling score for top 5 brands by company 2015–2018.**

based on their value sales, although they are also the top 10 companies in terms of volume sales. By selecting companies based on their global, rather than national, names, UK retailers were excluded from the analysis. This is a major limitation given that own-label brands from the top 3 UK retailers (Tesco, Sainsbury's and Asda) represented a total market share of 21% in 2018 [5]. While this study sets out a useful and important method for ranking companies in terms of healthiness of product portfolios, future studies should include retailers and a wider range of companies. This would give a more comprehensive picture of how food and drink companies and retailers in the UK are changing their products to meet public health targets. There are a number of data-driven limitations. The first is in relation to missing and imputed data. The values for seven nutrients (energy, saturated fat, total sugars, sodium, fibre, protein

and FNV content) are needed to calculate the FSA/Ofcom nutrient profile score of a product. 32% of the 13,371 products included in this study had missing values for fibre, and 67% products had insufficient ingredients information and composition detail to be able to calculate % FNV accurately. There was no difference in the proportion of missing values over time. Missing values were imputed with either a category and/or brand average. The high proportion of missing/imputed fibre and FNV was to be expected as the labelling of fibre on foods is not mandatory (unlike other macronutrients) [12] and the percentages for individual ingredients (i.e. FNV ingredients) only have to be stated when the product title includes an ingredient name, or when a claim about the amount of an ingredient has been made on the label [13].

To test what impact the imputed fibre data had on the results, a sensitivity analysis was conducted. 31% (n = 4186) of all products in the original dataset had imputed fibre values, and these were evenly distributed across the four years. For our sensitivity analysis, we adjusted the fibre content for these products to 0.0g/100g, with the FSA/Ofcom points awarded for fibre also then given 0, the lowest score possible. The number of products that were classified as healthy fell from 47% to 46% in 2018, and there were negligible changes in the total sales-weighted nutrient profiling score for 2018, which fell from 51 to 50.

For fruit, nut and vegetable (FNV) content, 8896 (67%) of included products had imputed values, although three-quarters of these (n = 6624) fell into categories that you would not expect to contain enough FNV to score one point: baked goods, confectionery, dairy, ice cream, savoury snacks, soft drinks, spreads and staples. To test what impact the imputed FNV data may have had on the results, the remaining 2272 products (baby food, breakfast cereals, ready meals, and sauces, dressings and condiments) had their %FNV adjusted to 0%. After this adjustment, 25% (n = 564) of the 2272 products saw a change in their final Ofcom score. The overall proportion of products classified as healthy in 2018 fell from 47% to 46%. The results were the same as those found with the fibre sensitivity analysis, with a similar group of products being affected by the lack of fibre and FNV values. These results suggest that while the missing fibre and FNV values is a weakness in the dataset, the interpretation of the data was unchanged, and it has not affected the overall results.

Data restrictions meant that time period covered changes between 2015 and 2018. Previous reformulation efforts made before 2015, for example as part of the salt reduction programme that began in 2006, will have been excluded. Using a wider historic time period may show that some companies who started reformulation efforts promptly have made more signficant changes than recorded here. Applying this method to datasets in multiple countries may offer insight into how companies are responding in countries with varying public health nutrition policies, for example voluntary reformulation targets in the UK compared to taxes on energy dense foods in Mexico [14] and mandatory warning labels in Chile [15].

The FSA/Ofcom nutrient profile model was used because it is designed for and used in the UK market and has been widely validated in terms of how its use may impact on dietary choices [16]. However, its original purpose was for the assessment of whether or not a product should be advertised to children, rather than to assess the nutritonal quality of a company's product portfolio and classifying products as healthy and unhealthy, as it was used here. It would be possible to conduct similar analyses using other nutrient profiling models such as Health Star Rating [17] and Nutri Score [18], though since all rely on changes in the underlying nutrient composition differences between scoring systems are likley to be modest.

We combined the distributions of food and drink products by using a linear transformation that matched the distributions at two points–the 33rd and 66th percentile. The selection of the two matching points was arbitrary. Matching at different points (e.g. the 25th and 75th percentiles) would have produced a different linear transformation and hence different scores for

drinks. This is an inevitable limitation associated with combining scores for companies with both food and drink profiles.

**Comparisons with other studies.**    There are a number of studies that have examined the nutrient content of foods sold in the UK over time. Previous studies have shown that voluntary salt reduction targets in the UK led to gradual and important changes in the salt content of foods between 2008–2011 [19,20], although a more recent report from Public Health England (PHE) suggests that only 28 of 52 of the 2017 salt reduction targets had been met in 2018 [4]. Two studies have shown that there were significant changes in the sugar content of soft drinks in the UK in context of the introduction of the Soft Drink Industry Levy [10,11]. The changes in the sugar content of soft drinks presented in these studies is in line with the results presented here, where the majority of the change in the volume sales of foods classified as healthy was driven by changes in the sugar content of soft drinks. Another study has also looked at the sugar content of foods between 2015 and 2018 and also presented findings by category and company [21].This study showed that 24 out of the top 50 companies (including retailers) in the UK had met Public Health England's 5% sugar reduction targets, and that companies have made limited progress towards meeting this voluntary policy. Public Health England have themselves published a series of reports that monitor progress being made towards their 20% sugar reduction targets using both sales and composition data [3]. For example, they have shown that there was a -2.9% reduction in the sugar content of foods between 2015 and 2018 [3]. A strength of our study is that it applies a nutrient profiling model, whereas these analyses are based on single nutrients and are therefore not directly comparable. However, they generally show that there has been mixed progress by the food industry towards public health goals.

INFORMAS (International Network on Food and Obesity/NCD Research, Monitoring and Action Support) have produced a series of company scorecards that rank the world's top 25 food companies, including supermarkets and quick-service restaurants, in a number of different areas, including product formulation [22]. While the scores are not based on quantitative analysis of the nutritional quality of companies' products, they are based on business practices and companies' commitments to nutrition-related policies, which is also important for monitoring food industry progress towards public health goals.

In 2019, the Access to Nutrition Initiative (ATNI) published its UK Product Profile [23]. It analysed the nutritional quality of 3069 products from the top five food categories of the world's top 18 manufacturers in 2016. The ATNI study also applied the HSR nutrient profiling model. Nine companies (excluding Premier Foods, a UK-only company) included here were also included in the ATNI index. ATNI found that 31% of products were classified as healthy enough to advertise to children, compared to 45% in 2016 here. 22% of sales were classified as healthy, as opposed to 55% in this study. These differences are likely to be accounted for by the fact that ATNI had a lower coverage (this study included 3438 products for 10 companies in 2016, compared to 3069 products for 18 companies for ATNI). The main advantage of this study over ATNI's UK Product Profile is that it includes four years' worth of data and therefore examines trends over time, whereas ATNI's study is a snapshot of a single year. The two studies are not directly comparable as the ATNI companies were defined at the global level, rather than UK level, and therefore the brands included under each company vary. However, the general ranking of the companies were similar between the two studies; Kraft Heinz and Danone were the two top scoring companies, and Nestlé, Mars and Mondelez were ranked at the bottom.

Another study similar to this one, conducted in India by Jones et al. 2017, used Euromonitor sales data and nutrition composition data for 943 products, collected from either the packet or company websites [24]. It applied the Health Star Rating (HSR) to analyse the nutritional quality of the top 11 packaged food manufacturers in India. The study found that the overall

healthiness of products was low and that only 17% of products were considered healthy [24]. This is lower than the 45% of products classified as healthy in this study in 2016. These differences are to be expected as the Indian study excluded products like staples (bread, pasta, rice), and used a different nutrient profiling model (HSR). Despite covering a very different market, it demonstrates that a high proportion of products sold by leading companies in other countries are also unhealthy, and that this problem is not isolated to the UK.

### Implications of research

This study shines a spotlight on the very small changes over time in the nutritional quality of food and drink products from the UKs largest food and beverage companies. While the proportion of volume sales increased from 44% to 53% over time, this change was entirely down to increased volume sales of bottled water, low/no calorie drinks and high-scoring fruit juices. The brands that saw the biggest changes to their scores over time were soft drinks. Once soft drinks were removed, the total volume sales of foods classified as healthy dropped to just 6% in 2018, down from 7% in 2015. This strongly suggests that PHE's reformulation targets for sugar, salt and calories have not had a substantive impact on the nutritional quality of foods.

This method of ranking food and drink companies based on the nutritional quality of their product portfolios could be used to benchmark companies as a tool for 'healthier' impact investment. There is an increasing interest by investment banks and other financial organisations to assess what impact food companies are having on public health and how responsible their business practices are (known as impact investment) [25]. This has already been done in part by ATNI in collaboration with Shared Action [26] and INFORMAS [22].

Transparent monitoring of this kind also allows for greater consumer understanding of the work that is, or is not, being undertaken by companies. There is some evidence that pressure from the social environment is a factor influencing corporate behaviour [27], and public benchmarking exercises may increase pressure on companies to make meaningful change.

### Conclusion

This study has demonstrated that it is feasible to monitor overall healthiness of company product portfolios over time. It shows that companies have made little change to the nutritional quality of their product portfolios, despite a few individual brand success stories, a factor which needs to be considered by policy makers when reviewing the current focus on single-nutrient reformulation programmes. Implementing a transparent monitoring and evaluation system such as this, would allow for targeted work with the companies to drive improvements in public health nutrition.

### Author Contributions

**Conceptualization:** Lauren Kate Bandy, Susan Jebb, Mike Rayner.

**Data curation:** Lauren Kate Bandy, Sven Hollowell, Richard Harrington.

**Formal analysis:** Lauren Kate Bandy, Sven Hollowell, Peter Scarborough.

**Investigation:** Lauren Kate Bandy, Peter Scarborough, Mike Rayner.

**Methodology:** Lauren Kate Bandy, Sven Hollowell, Richard Harrington, Peter Scarborough, Mike Rayner.

**Project administration:** Lauren Kate Bandy.

**Software:** Lauren Kate Bandy, Sven Hollowell.

**Supervision:** Richard Harrington, Peter Scarborough, Susan Jebb, Mike Rayner.

**Validation:** Lauren Kate Bandy, Sven Hollowell.

**Visualization:** Peter Scarborough.

**Writing – original draft:** Lauren Kate Bandy, Peter Scarborough, Susan Jebb, Mike Rayner.

**Writing – review & editing:** Lauren Kate Bandy, Sven Hollowell, Richard Harrington, Peter Scarborough, Susan Jebb, Mike Rayner.

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
