## [Decision Letter · Decision Letter 0]

15 Apr 2021

PONE-D-21-01154

Assessing the healthiness of UK food companies’ product portfolios using food sales and nutrient composition data.

PLOS ONE

Dear Dr. Bandy,

Thank you for submitting your manuscript to PLOS ONE. After careful consideration, we feel that it has merit but does not fully meet PLOS ONE’s publication criteria as it currently stands. Therefore, we invite you to submit a revised version of the manuscript that addresses the points raised during the review process.

We look forward to receiving your revised manuscript.

Kind regards,

Jane Anne Scott, PhD, MPH Grad Dip Dietetics, BSc

Academic Editor

PLOS ONE

Additional Editor Comments:

Line 125 Suggest rewording.’ Products that scored below relevant scores were classified as unhealthy.’ As it was not just foods but also beverages.

Line 145 Did the exclusion of infant formulas include the exclusion of toddler formulas that are not officially classified as breastmilk substitutes?

Line 322 should read 3069 products

Line 296 please provide references for the other nutrient profiling models referred to i.e. Health Stare Rating and Nutriscore

Journal Requirements:

Reviewers' comments:

Reviewer's Responses to Questions

**Comments to the Author**

1. Is the manuscript technically sound, and do the data support the conclusions?

Reviewer #1: Yes

Reviewer #2: Yes

2. Has the statistical analysis been performed appropriately and rigorously? 

Reviewer #1: Yes

Reviewer #2: Yes

3. Have the authors made all data underlying the findings in their manuscript fully available?

Reviewer #1: No

Reviewer #2: No

4. Is the manuscript presented in an intelligible fashion and written in standard English?

Reviewer #1: Yes

Reviewer #2: Yes

5. Review Comments to the Author

Reviewer #1: This was a nicely-written, simple yet effectively presented study examining changes in the nutrient profile score of food and beverage products from 10 major manufacturers in the UK. The study is certainly of interest and timing-wise is important given the attention of late to marketing to children in the UK and EU.

Overall, my main comments and suggestions for improvement relate to the authors interpretation of the results and to the comparison with existing studies. I have split my comments into major versus minor. All I think are easily actionable and I would recommend they all be addressed prior to this being accepted for publication.

Major

1. What about retailers? In the UK retailer own brands/private label dominate the food supply. This is a huge limitation of the paper as it stands and so interpretation of results needs to be made more cautiously given that the largest part of the market has not been included. Authors mention this briefly as a limitation, however I recommend some commentary on what % of the market the retailers represent, and whether any specific retailer has % sales higher than one of the top 10 companies? I personally don’t know is this is the case but it should be mentioned for sure. Perhaps a rewording to say you are looking at global food and beverage companies within the context of the UK market? Only Premier Foods is a UK company, all others in the top 10 are global companies.

2. How did imputation differ between 2015 and 2018 – the authors mention substantial imputation was needed, yet my sense would be that if this differed between years, then some kind of sensitivity analysis would be warranted to see whether changes could be attributed to estimated nutrient values rather than true change. I appreciate sensitivity analyses was done with and without imputation, but understanding how this affected each year’s dataset is also important.

3. I caution the authors to revise the text in the sections comparing results to previous studies to better clarify the likely reasons for differences in results, rather than dismiss the results of the previous studies for being flawed – as it is written it comes across as if the present study is much better/more accurate than previous studies, which I do not believe is the case – each study uses different methodologies (and different companies in some cases) and hence would be completely expected to yield different results (especially when comparing India to the UK – these are VASTLY different packaged food supplies!). PHE I am sure would also include retailers? This would give vastly different results than looking only at global manufacturers. Have the authors looked at previous studies examining changes in the nutritional content of UK foods – there are certainly studies out there on sodium, and likely others too. The discussion section was definitely “thin” on references and seemed to focus on a small number of studies alone.

Minor

1. Line 69/70 – “in 2018” is repeated

2. Line 124 – why? Is this based on something? Need to justify these cut-offs

3. Line 148 – which program was used

4. Line 173-174 – remove the sentence about soft drinks as it was a non-significant result. Or else change the wording to say there was no difference.

5. Line 225 – should be “Coco-Pops” for consistency

6. Line 280 – spelling error – drink

7. Small thing – should be “PepsiCo” with the C capitalised

Reviewer #2: Thank you for the opportunity to review this manuscript – assessing the healthiness of UK food companies’ product portfolios using food sales and nutrient composition data. Overall, the manuscript is written well, and the methods applied are thorough. I have a number of comments that could further improve the manuscript.

Abstract

- in methods, the threshold of nutrient profile score for classifying products as healthy vs. unhealthy hasn’t been provided. Describing briefly the threshold will help readers understand better the results.

- Line 48, is it ‘only 6 brands among the 10 companies improved …’ ?

- Lines 46 and 47 ‘… the number of products classified as healthy (46% to 48%)…’ the statement says ‘the number’ but results are in ‘%’

- Line 49, proportion of total volume sales (classed?) classified as healthier increased from 44% to 51% …; but in line 52, the proportion of foods classified as healthy decreased from 7% to 6%... is confusing. It’s important to be consistent which means the decrease should be the ‘proportion of total volume sales classified as healthier’. Moreover, the use of healthier in some places and healthy in other places is confusing. The authors may choose to use ‘healthy’ whenever ‘healthier’ refers to ‘healthy’ as a binary variable created using the threshold of ≥62 for foods and ≥66 nutrient profiling score for drinks.

Methods

- Line 108, ‘… data were automatically matched based on product name, brand name, category and year, …’ needs to be explained on how the authors created the identifier variable in the sales database and nutrient composition database. By identifier variable I mean the variable that was used to link (match) the two databases.

- Line 111, ‘… The matching code was adjusted, …’ The authors please explain how this adjustment was done.

- Line 142, ‘… weighted mean nutrient profiling scores were calculated for each company and brand, …’. It is well written here; however, in figure3 and other figures it says ‘sales-weighted score’. The authors may choose to clarity it here or may choose to change ‘Variable calculated’ to ‘Sales-weighted score’ in line 140.

- Line 146, … Infant formulas (i.e., breast milk substitutes) were excluded. Were there other products to be excluded such as products not required to display a Nutrition Information Panel (e.g. tea, unflavored coffee, artificial sweeteners, chewing and bubble gums, salt, flour, corn flour, vinegar, herbs and spices, pepper, baking soda, baking powder, tartaric acid, citric acid, cooking ingredients, ice, curry powder, yeast, bicarbonate of soda. Other foods such as special products (baby foods, protein bars, protein powders, and fitness or diet products), and alcoholic beverages may also be excluded. If these products were not available in the database, then the authors may mention about them.

Results

- In Table 1. What does ‘per capita per day’ refer to?

- Line 173, does ‘healthier’ mean ‘healthy’? based on the threshold described in line 125

- Line 173 says ‘… increase from 44% to 51% …’ but in line 174 after removing soft drinks from the analysis, the proportion of volume sales classified as healthier decreased from 7% to 6% (p=0.33). I assume that 7% refers to 51% minus 44%, and 6% refers to 50% minus 44%. I am not sure about it, and it think its needs to be clarified here.

- I am unsure whether ‘bottled water’ should be excluded from the analysis, especially it may favor manufacturing of more healthy beverages by Coca-Cola, and PepsiCo.

- Results in page 8 showing significant improvement in nutrient profiling score for soft drink brands give the impression that these brands (i.e., San Pellegrino, Fanta, Volvic, Tropicana, Kellogg) are doing good in manufacturing more healthy beverages. I’m not sure whether inclusion of ‘bottled water’ in the analysis has affected the results or whether reductions in salt, sugar and energy have led to these results.

Discussion

- Line 238 ‘… foods and drinks available, I understand the authors studied sales; but did the authors examine availability too?

- Line 240 ‘… responding to public health calls to improve …’ is not specific. The authors may choose to write ‘… responding to the voluntary reformulation initiatives to improve’

- Lines 246 to 274 mainly describe and discuss ‘missing values’ and ‘imputation’. They can be divided between methods, results and discussion. If the authors think the results related to missing values, imputation and sensitivity analysis can distract the readers, the authors may choose to provide them as an appendix.

- Line 261, … (-0.5%) refers to what?

- Line 288, please cite the references in ‘… energy dense foods in Mexico [reference?] and mandatory warning labels in Chile [reference?]’.

One of the strengths of this study can be use of nutrient profiling score that the authors may want to highlight it in the discussion. In this regard lines 312 and 313 say … these analyses consider only single nutrients and do not apply a nutrient profile model …’ but these are not emphasizing on this strength of this study.

6. PLOS authors have the option to publish the peer review history of their article (what does this mean?). If published, this will include your full peer review and any attached files.

Reviewer #1: No

Reviewer #2: **Yes: **Essa Tawfiq

---

## [Author Response · Author response to Decision Letter 0]

24 Jun 2021

[PONE-D-21-01154] Reviewer response letter

Editor Comments:

Line 125 Suggest rewording.’ Products that scored below relevant scores were classified as unhealthy.’ As it was not just foods but also beverages.

Thanks, this has been corrected

Line 145 Did the exclusion of infant formulas include the exclusion of toddler formulas that are not officially classified as breastmilk substitutes?

Line 151 now reads: “Milk formulas for infants, toddlers and children were excluded.” We hope this has clarified this.

Line 322 should read 3069 product

Thanks, this has been corrected (line 349)

Line 296 please provide references for the other nutrient profiling models referred to i.e. Health Star Rating and Nutriscore

Thanks, these have been added (line 307)

Journal Requirements:

Funding statement: 

LB, SH and MR are funded by the Nuffield Department of Population Health, University of Oxford. PS is funded by a British Heart Foundation Intermediate Basic Science Research Fellowship (FS/15/34/31656). All authors are part of the National Institute for Health Research (NIHR) Oxford Biomedical Research Centre (BRC). SJ is also funded by the NIHR Collaboration for Leadership in Applied Health Research and Care Oxford at Oxford Health NHS Foundation Trust and is an NIHR senior investigator. The funders had no role in study design, data collection and analysis, decision to publish, or preparation of the manuscript

Please see the updated data availability statement below:

This study used data from two commercial sources. The sales data was accessed under licence from Euromonitor International (https://www.euromonitor.com/packaged-food) via the Bodleian Library, University of Oxford, using Euromonitor’s database portal Passport GMID. The product information dataset, including nutrition composition data, was purchased for the purpose of the lead author’s DPhil research project from Edge by Ascential (https://www.ascentialedge.com/our-solutions). Due to licencing restrictions, the Euromonitor and Edge by Ascential datasets can only be requested under licence for the purpose of verification and replication of study’s findings via the research group’s Data Access Committee (contact: Trisha Gordon foodDBaccess@ndph.ox.ac.uk). Further use of these datasets must be negotiated with the data owners (Euromonitor contact: Ashton Moses - passport.support@euromonitor.com, Edge by Ascential contact: David Beech - info@ascentialedge.com).

Reviewer #1: This was a nicely-written, simple yet effectively presented study examining changes in the nutrient profile score of food and beverage products from 10 major manufacturers in the UK. The study is certainly of interest and timing-wise is important given the attention of late to marketing to children in the UK and EU.

Overall, my main comments and suggestions for improvement relate to the authors interpretation of the results and to the comparison with existing studies. I have split my comments into major versus minor. All I think are easily actionable and I would recommend they all be addressed prior to this being accepted for publication.

Major

1. What about retailers? In the UK retailer own brands/private label dominate the food supply. This is a huge limitation of the paper as it stands and so interpretation of results needs to be made more cautiously given that the largest part of the market has not been included. Authors mention this briefly as a limitation, however I recommend some commentary on what % of the market the retailers represent, and whether any specific retailer has % sales higher than one of the top 10 companies? I personally don’t know is this is the case but it should be mentioned for sure. Perhaps a rewording to say you are looking at global food and beverage companies within the context of the UK market? Only Premier Foods is a UK company, all others in the top 10 are global companies.

We thank the reviewer for their comment and agree that the exclusion of retailers from the list of companies is a central limitation. Euromonitor classifies company names in two ways, by Global Brand Owner (GBO – the ultimate global owner of the brand) and National Brand Owner (NBO – the UK registered producer (including under licence) or distributor of the brand). We selected the top 10 companies based on GBO rather than NBO. Under GBO, all retailers are grouped under the umbrella term ‘Private Label’ and individual retailer names are not given. When the top 10 companies in the UK are viewed by NBO, the individual retailers are named and 5 out of 10 of the top 10 companies are indeed retailers. Top 10 by GBO was chosen for the main reason that we used composition data from Brand View, which only provided data for three of these five retailers (Tesco, Sainsbury’s and Asda), and so we wouldn’t be able to compare all 5 retailers that featured in the top 10 companies (Morrisons and Marks & Spencer being the other two). We also thought global companies might be more recognisable for a global audience. 

We have edited the methods section to make it clear that the top 10 UK companies were selected based on global name (line xx) and the strengths and limitations section of the discussion (line xx) now reads: “Only 10 companies, based on global company name, were included in the analysis, which represented 24% of total value sales in the UK in 2018 [5]. These companies were selected based on their value sales, although they are also the top 10 companies in terms of volume sales. By selecting companies based on their global, rather than national, names, UK retailers were excluded from the analysis. This is a major limitation given that own-label brands from the top 3 UK retailers (Tesco, Sainsbury’s and Asda) represented a total market share of 21% in 2018 [5]. While this study sets out a useful and important method for ranking companies in terms of healthiness of product portfolios, future studies should include retailers and a wider range of companies. This would give a more comprehensive picture of how food and drink companies and retailers in the UK are changing their products to meet public health targets.”

Our research group has developed a tool that collects product information data (including nutrition composition and ingredients) from 10 supermarket websites in the UK and therefore we are planning to undertake a similar study that will score and rank UK retailers based on the healthiness of their own-label products. We anticipate that this will be a complimentary paper to this one, and hope that the corrections we have made here have highlighted the limitations in excluding retailers from our analysis.

2. How did imputation differ between 2015 and 2018 – the authors mention substantial imputation was needed, yet my sense would be that if this differed between years, then some kind of sensitivity analysis would be warranted to see whether changes could be attributed to estimated nutrient values rather than true change. I appreciate sensitivity analyses was done with and without imputation, but understanding how this affected each year’s dataset is also important.

There was no difference in the proportion of values that were imputed over time, and a line has been added to the methods section (line 263) to make this clear to the reader. The proportion of imputed FNV values was 29% (n=994) in 2015 and 30% (n=1010) in 2018, and the proportion of imputed fibre values was 68% (n=2336) in 2015 and 71% (n=2132) in 2018. Given the lack of difference over time, a sensitivity analysis to test the impact of imputed data on each year’s dataset was not deemed necessary.

3. I caution the authors to revise the text in the sections comparing results to previous studies to better clarify the likely reasons for differences in results, rather than dismiss the results of the previous studies for being flawed – as it is written it comes across as if the present study is much better/more accurate than previous studies, which I do not believe is the case – each study uses different methodologies (and different companies in some cases) and hence would be completely expected to yield different results (especially when comparing India to the UK – these are VASTLY different packaged food supplies!). PHE I am sure would also include retailers? This would give vastly different results than looking only at global manufacturers. Have the authors looked at previous studies examining changes in the nutritional content of UK foods – there are certainly studies out there on sodium, and likely others too. The discussion section was definitely “thin” on references and seemed to focus on a small number of studies alone.

We recognise that the references in the comparison with previous studies section were light, and have added six studies that look at the sugar and salt content of foods over time. We have also tried to rephrase this section to ensure it’s not overly critical and focuses on how the comparative study adds important context to the findings we have set out. The “Comparison with other studies” section (lines 319-375) now reads: 

“There are a number of studies that have examined the nutrient content of foods sold in the UK over time. Previous studies have shown that voluntary salt reduction targets in the UK led to gradual and important changes in the salt content of foods between 2008-2011 [19][20], although a more recent report from Public Health England (PHE) suggests that only 28 of 52 of the 2017 salt reduction targets had been met in 2018 [4]. Two studies have shown that there were significant changes in the sugar content of soft drinks in the UK in context of the introduction of the Soft Drink Industry Levy [10][11]. The changes in the sugar content of soft drinks presented in these studies is in line with the results presented here, where the majority of the change in the volume sales of foods classified as healthy was driven by changes in the sugar content of soft drinks. Another study has also looked at the sugar content of foods between 2015 and 2018 and also presented findings by category and company [21].This study showed that 24 out of the top 50 companies (including retailers) in the UK had met Public Health England’s 5% sugar reduction targets, and that companies have made limited progress towards meeting this voluntary policy. Public Health England have themselves published a series of reports that monitor progress being made towards their 20% sugar reduction targets using both sales and composition data [3]. For example, they have shown that there was a -2.9% reduction in the sugar content of foods between 2015 and 2018 [3]. A strength of our study is that it applies a nutrient profiling model, whereas these analyses are based on single nutrients and are therefore not directly comparable. However, they generally show that there has been mixed progress by the food industry towards public health goals. 

INFORMAS (International Network on Food and Obesity/NCD Research, Monitoring and Action Support) have produced a series of company scorecards that rank the world’s top 25 food companies, including supermarkets and quick-service restaurants, in a number of different areas, including product formulation [22]. While the scores are not based on quantitative analysis of the nutritional quality of companies’ products, they are based on business practices and companies’ commitments to nutrition-related policies, which is also important for monitoring food industry progress towards public health goals.

In 2019, the Access to Nutrition Initiative (ATNI) published its UK Product Profile [23]. It analysed the nutritional quality of 3069 products from the top five food categories of the world’s top 18 manufacturers in 2016. The ATNI study also applied the HSR nutrient profiling model. Nine companies (excluding Premier Foods, a UK-only company) included here were also included in the ATNI index. ATNI found that 31% of products were classified as healthy enough to advertise to children, compared to 45% in 2016 here. 22% of sales were classified as healthy, as opposed to 55% in this study. These differences are likely to be accounted for by the fact that ATNI had a lower coverage (this study included 3438 products for 10 companies in 2016, compared to 3069 products for 18 companies for ATNI). The main advantage of this study over ATNI’s UK Product Profile is that it includes four years’ worth of data and therefore examines trends over time, whereas ATNI’s study is a snapshot of a single year. The two studies are not directly comparable as the ATNI companies were defined at the global level, rather than UK level, and therefore the brands included under each company vary. However, the general ranking of the companies were similar between the two studies; Kraft Heinz and Danone were the two top scoring companies, and Nestlé, Mars and Mondelez were ranked at the bottom. 

Another study similar to this one, conducted in India by Jones et al. 2017, used Euromonitor sales data and nutrition composition data for 943 products, collected from either the packet or company websites [24]. It applied the Health Star Rating (HSR) to analyse the nutritional quality of the top 11 packaged food manufacturers in India. The study found that the overall healthiness of products was low and that only 17% of products were considered healthy [24]. This is lower than the 45% of products classified as healthy in this study in 2016. These differences are to be expected as the Indian study excluded products like staples (bread, pasta, rice), and used a different nutrient profiling model (HSR). Despite covering a very different market, it demonstrates that a high proportion of products sold by leading companies in other countries are also unhealthy, and that this problem is not isolated to the UK.” 

Minor

1. Line 69/70 – “in 2018” is repeated

Thanks, this has been corrected.

2. Line 124 – why? Is this based on something? Need to justify these cut-offs

This section has been rephrased to make this clearer and line 133-135 now reads: “After the scale conversion and linear adjustment, the thresholds for products to be considered healthy according to the FSA/Ofcom nutrient profile model were 62 or more for foods and 66 or more for drinks.”

3. Line 148 – which program was used

Analyses were conducted in R and we have added corrections to line 148 and 155 to make this clear

4. Line 173-174 – remove the sentence about soft drinks as it was a non-significant result. Or else change the wording to say there was no difference.

We have kept this line, as while the change over time wasn’t significant, removing soft drinks from the results did result in a very different result in terms of proportion of volume sales classified as health (51% in 2018 compared to 6% once soft drinks were removed). This result is important to the narrative that the majority of any change seen was from soft drinks, where mandatory policy was applied, compared to voluntary reformulation targets for packaged foods. 

5. Line 225 – should be “Coco-Pops” for consistency

Thanks, this has been corrected

6. Line 280 – spelling error – drink

Thanks, this has been corrected

7. Small thing – should be “PepsiCo” with the C capitalised

Thanks, this has been corrected

Reviewer #2: Thank you for the opportunity to review this manuscript – assessing the healthiness of UK food companies’ product portfolios using food sales and nutrient composition data. Overall, the manuscript is written well, and the methods applied are thorough. I have a number of comments that could further improve the manuscript.

Abstract

- in methods, the threshold of nutrient profile score for classifying products as healthy vs. unhealthy hasn’t been provided. Describing briefly the threshold will help readers understand better the results.

Thanks, the thresholds have been added as suggested and line 141 onwards reads: “The mean nutrient profiling score, on a scale of 1-100 with thresholds for healthy products being 62 for foods and 68 for drinks, was used to rank companies and food categories between 2015 and 2018..”

- Line 48, is it ‘only 6 brands among the 10 companies improved …’ ?

Thanks, this has been corrected

- Lines 46 and 47 ‘… the number of products classified as healthy (46% to 48%)…’ the statement says ‘the number’ but results are in ‘%’

Thanks, this has been corrected to ‘proportion’ 

- Line 49, proportion of total volume sales (classed?) classified as healthier increased from 44% to 51% …; but in line 52, the proportion of foods classified as healthy decreased from 7% to 6%... is confusing. It’s important to be consistent which means the decrease should be the ‘proportion of total volume sales classified as healthier’. Moreover, the use of healthier in some places and healthy in other places is confusing. The authors may choose to use ‘healthy’ whenever ‘healthier’ refers to ‘healthy’ as a binary variable created using the threshold of ≥62 for foods and ≥66 nutrient profiling score for drinks.

Thanks for your comment – we have changed the word ‘healthier’ to ‘healthy’ throughout the manuscript wherever we refer to a binary variable created using the thresholds for food and drink scores.

Methods

- Line 108, ‘… data were automatically matched based on product name, brand name, category and year, …’ needs to be explained on how the authors created the identifier variable in the sales database and nutrient composition database. By identifier variable I mean the variable that was used to link (match) the two databases.

This section has now been updated as suggested and reads: “The sales data and nutrition composition data were automatically matched in Python based on three identifier variables that were present in both databases: brand name, category and year.” (lines 109-111)

- Line 111, ‘… The matching code was adjusted, …’ The authors please explain how this adjustment was done.

This section has now been updated as suggested and reads: “A 10% random sample of brands checked manually for any errors. Of the 20 brands checked, 4 brands were identified as pairing with the correct brand name but incorrect category. All 4 of these errors were brands that appeared in more than one category (e.g. Cadbury is present five categories, including baked goods and confectionery). The matching code was adjusted so that it first paired based on matching categories, and then brand names, and no errors were identified after further checks.” (Lines 111-117) 

- Line 142, ‘… weighted mean nutrient profiling scores were calculated for each company and brand, …’. It is well written here; however, in figure3 and other figures it says ‘sales-weighted score’. The authors may choose to clarity it here or may choose to change ‘Variable calculated’ to ‘Sales-weighted score’ in line 140.

We agree that ‘sales-weighted mean nutrient profiling scores’ is clearer, but it was too long to use for the title of the figure key (character limit), therefore we have kept the figure key title the same, but added the following line (line 146) to the methods section: “The total value (£ millions) and volume of food and soft drinks (tonnes) and the sales weighted mean nutrient profiling score (referred to in figure labels as sales-weighted score) were calculated in R for each company and brand, both overall and by category.”

- Line 146, … Infant formulas (i.e., breast milk substitutes) were excluded. Were there other products to be excluded such as products not required to display a Nutrition Information Panel (e.g. tea, unflavored coffee, artificial sweeteners, chewing and bubble gums, salt, flour, corn flour, vinegar, herbs and spices, pepper, baking soda, baking powder, tartaric acid, citric acid, cooking ingredients, ice, curry powder, yeast, bicarbonate of soda. Other foods such as special products (baby foods, protein bars, protein powders, and fitness or diet products), and alcoholic beverages may also be excluded. If these products were not available in the database, then the authors may mention about them.

We only selected products manufactured by the 10 companies presented in the studies. The only products we excluded were infant formulas and bubble and chewing gum (now updated in line 151).

Results

- In Table 1. What does ‘per capita per day’ refer to?

This refers to the equivalent value spend per person and the column heading as been updated to make this clearer.

- Line 173, does ‘healthier’ mean ‘healthy’? based on the threshold described in line 125

Thanks for your comment – we have changed the word ‘healthier’ to ‘healthy’ throughout the manuscript wherever we refer to a binary variable created using the thresholds for food and drink scores.

- Line 173 says ‘… increase from 44% to 51% …’ but in line 174 after removing soft drinks from the 

analysis, the proportion of volume sales classified as healthier decreased from 7% to 6% (p=0.33). I assume that 7% refers to 51% minus 44%, and 6% refers to 50% minus 44%. I am not sure about it, and it think its needs to be clarified here.

We have added the years to this sentence and hope that this clarifies the meaning. Lines 179-181 now read: “Once soft drinks were removed, the proportion of volume sales that were classified as healthy decreased from 7% in 2015 to 6% in 2018 (p = 0.33).”

- I am unsure whether ‘bottled water’ should be excluded from the analysis, especially it may favor manufacturing of more healthy beverages by Coca-Cola, and PepsiCo.

We decided to include bottled water in the analysis, as shifting sales from sugary drinks to bottled water would be considered to be a positive industry health behaviour. Additionally, we also adjusted the soft drink scores to make them directly comparable to food scores, so that soft drink companies are not favoured.

- Results in page 8 showing significant improvement in nutrient profiling score for soft drink brands give the impression that these brands (i.e., San Pellegrino, Fanta, Volvic, Tropicana, Kellogg) are doing good in manufacturing more healthy beverages. I’m not sure whether inclusion of ‘bottled water’ in the analysis has affected the results or whether reductions in salt, sugar and energy have led to these results.

San Pellegrino and Volvic are bottled water brands and so increased sales of these will have affected the results. Fanta, Kellogg and Tropicana will not be affected by changes to bottled water sales. Please see above regarding the reason for including bottled water in the anaylsis.

Discussion

- Line 238 ‘… foods and drinks available, I understand the authors studied sales; but did the authors examine availability too?

Where results are presented as number of individual products classified as healthy (i.e. not weighted by sales), then this is a marker of availability.

- Line 240 ‘… responding to public health calls to improve …’ is not specific. The authors may choose to write ‘… responding to the voluntary reformulation initiatives to improve’

Thanks, this has been updated as recommended.

- Lines 246 to 274 mainly describe and discuss ‘missing values’ and ‘imputation’. They can be divided between methods, results and discussion. If the authors think the results related to missing values, imputation and sensitivity analysis can distract the readers, the authors may choose to provide them as an appendix.

Thanks for your comment - we decided that even with the missing values/imputation sections in the main body of the text it was succinct enough, and therefore we have not moved anything to an appendix

- Line 261, … (-0.5%) refers to what?

Thanks, this appears to be a mistake and has been removed.

- Line 288, please cite the references in ‘… energy dense foods in Mexico [reference?] and mandatory warning labels in Chile [reference?]’.

Thanks, these references have been added (line 298).

One of the strengths of this study can be use of nutrient profiling score that the authors may want to highlight it in the discussion. In this regard lines 312 and 313 say … these analyses consider only single nutrients and do not apply a nutrient profile model …’ but these are not emphasizing on this strength of this study.

Thanks, we have updated this line as recommended and lines 336-339 now read: “A strength of our study is that it applies a nutrient profiling model, whereas these analyses are based on single nutrients and are therefore not directly comparable. However, they generally show that there has been mixed progress by the food industry towards public health goals.”

---

## [Editor Report · Decision Letter 1]

5 Jul 2021

Assessing the healthiness of UK food companies’ product portfolios using food sales and nutrient composition data.

PONE-D-21-01154R1

Dear Dr. Bandy,

We’re pleased to inform you that your manuscript has been judged scientifically suitable for publication and will be formally accepted for publication once it meets all outstanding technical requirements.

Kind regards,

Jane Anne Scott, PhD, MPH Grad Dip Dietetics, BSc

Academic Editor

PLOS ONE
---

## [Editor Report · Acceptance letter]

9 Jul 2021

PONE-D-21-01154R1 

Assessing the healthiness of UK food companies’ product portfolios using food sales and nutrient composition data. 

Dear Dr. Bandy:

I'm pleased to inform you that your manuscript has been deemed suitable for publication in PLOS ONE. Congratulations! Your manuscript is now with our production department. 

Kind regards, 

on behalf of

Dr. Jane Anne Scott 

Academic Editor

PLOS ONE